# The tardigrade *Hypsibius exemplaris* has the active mitochondrial alternative oxidase that could be studied at animal organismal level

Daria Wojciechowska[1,2], Milena Roszkowska[2], Łukasz Kaczmarek[3], Wiesława Jarmuszkiewicz[2], Andonis Karachitos[2], Hanna Kmita[2]*

**1** Faculty of Physics, Department of Macromolecular Physics, Adam Mickiewicz University, Uniwersytetu Poznańskiego, Poznań, Poland, **2** Faculty of Biology, Department of Bioenergetics, Adam Mickiewicz University, Uniwersytetu Poznańskiego, Poznań, Poland, **3** Department of Animal Taxonomy and Ecology, Adam Mickiewicz University, Uniwersytetu Poznańskiego, Poznań, Poland

\* kmita@amu.edu.pl

**Data Availability Statement:** All relevant data are within the manuscript and its Supporting Information files.

## Abstract

Mitochondrial alternative oxidase (AOX) is predicted to be present in mitochondria of several invertebrate taxa including tardigrades. Independently of the reason concerning the enzyme occurrence in animal mitochondria, expression of AOX in human mitochondria is regarded as a potential therapeutic strategy. Till now, relevant data were obtained due to heterologous AOX expression in cells and animals without natively expressed AOX. Application of animals natively expressing AOX could importantly contribute to the research. Thus, we decided to investigate AOX activity in intact specimens of the tardigrade *Hypsibius exemplaris*. We observed that *H. exemplaris* specimens' tolerance to the blockage of the mitochondrial respiratory chain (MRC) cytochrome pathway was diminished in the presence of AOX inhibitor and the inhibitor-sensitive respiration enabled the tardigrade respiration under condition of the blockage. Importantly, these observations correlated with relevant changes of the mitochondrial inner membrane potential ($\Delta\psi$) detected in intact animals. Moreover, detection of AOX at protein level required the MRC cytochrome pathway blockage. Overall, we demonstrated that AOX activity in tardigrades can be monitored by the animals' behavior observation as well as by measurement of intact specimens' whole-body respiration and $\Delta\psi$. Furthermore, it is also possible to check the impact of the MRC cytochrome pathway blockage on AOX level as well as AOX inhibition in the absence of the blockage on animal functioning. Thus, *H. exemplaris* could be consider as a whole-animal model suitable to study AOX.

## Introduction

Currently, it is suggested that the mitochondrial respiratory chain (MRC) of many invertebrates contains the alternative oxidase (AOX), an enzyme that provides a secondary oxidative pathway to the classical cytochrome pathway (e.g. [1]) but till now, natively expressed AOX level and activity have not been detected simultaneously in intact animals. Since the enzyme is

**Funding:** These studies were supported by the research grant of National Science Centre, Poland, NCN 2016/21/B/NZ4/00131.

**Competing interests:** The authors have declared that no competing interests exist.

**Abbreviations:** AA, antimycin A; AOX, mitochondrial alternative oxidase; BHAM, benzohydroxamic acid; FCCP, carbonyl cyanide 4-(trifluoromethoxy) phenylhydrazone; KCN, potassium cyanide; MRC, mitochondrial respiratory chain; TMRM, tetramethylrhodamine methyl ester.

not present in mitochondria of vertebrates, it is hypothesized that AOX-based respiration was critical to the evolution of animals by enabling oxidative metabolism during the transition to a fully oxygenated Earth atmosphere [2]. Thus, AOX may be treated as an adaptation to oxygen shortage which, in present-day organisms, may be caused by MRC deficiency or blockage. This in turn implies AOX application in treatment of mitochondrial diseases. As summarized by Saari et al. [3], the diseases may range from primary mitochondriopathies to common disease entities where mitochondrial disruption is due to ischemia/reperfusion injury, oxidative or proteotoxic stress, toxic damage or other external causes. However, the relevant data has been obtained by heterologous AOX expression in cells and animals that do not have native AOX, although it appears to be essential for understanding of AOX contribution to animal physiology and consecutive development of AOX-based therapeutic strategies. The AOX activity can be understood by detailed investigation of effects imposed by heterologous AOX expression on physiological responses of model organisms under stressful environmental conditions [3]. However, the research could be greatly simplified by application (as a model) of animals natively expressing AOX.

According to the definition, AOX is the mitochondrial inner membrane enzyme introducing a branch into the canonical animal MRC formed by four main multi-subunit complexes numbered from I to IV (Fig 1). The branching occurs before the MRC complex III, at the ubiquinone/ubiquinol pool, and results in transferring of electrons to oxygen with sustained proton pumping by the MRC complex I, but without proton pumping by the MRC complexes III and IV. The transfer is antimycin A (AA)- and cyanide-insensitive because AOX is not inhibited by AA and cyanides which are frequently used as inhibitors of the MRC complexes III and IV, i.e., ubiquinol–cytochrome c reductase and cytochrome c oxidase, respectively (e.g. [1,4–8]). Consequently, proton-pumping by MRC is confined to complex I which results in cyanide- and AA-resistant respiration. Since proton gradient, formed due to proton pumping, is required for ATP synthesis, this enables fine tuning of ATP synthesis as well as modulation of reactive oxygen species (ROS) and calcium ion levels (e.g. [1]). Resultantly, AOX is regarded to provide metabolic plasticity being useful for adaptation to variable biotic and abiotic stress

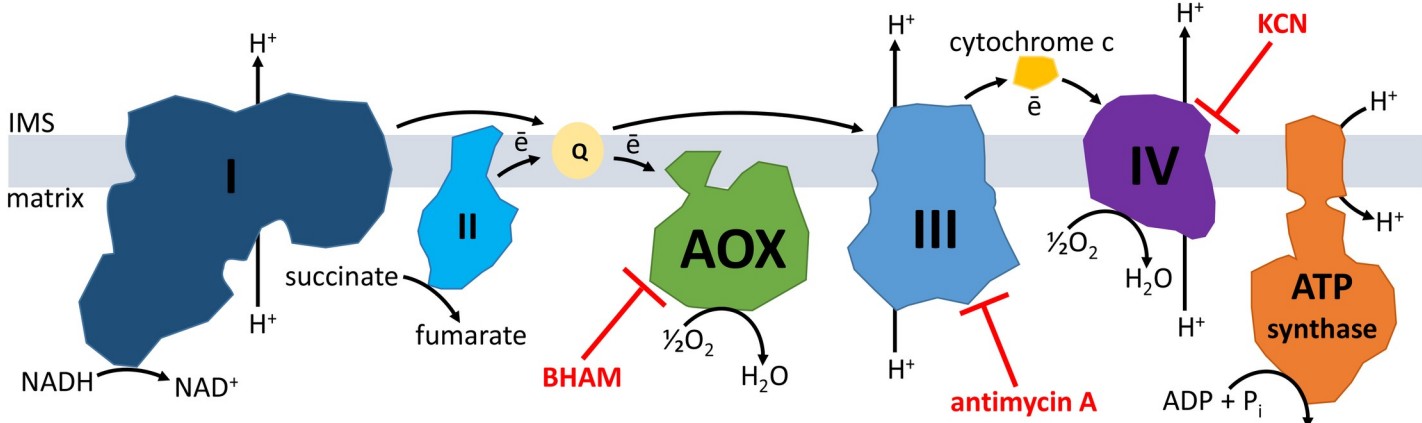

**Fig 1. Schematic representation of the respiratory chain in animal mitochondria.** The mitochondrial respiratory chain (MRC) in the inner mitochondrial membrane is formed by four main multi-subunit complexes numbered from I to IV. The presence of mitochondrial alternative oxidase (AOX) is postulated for invertebrates with exception of insects and lancelets. Electrons released due to oxidation of metabolites are transported by MRC to oxygen and this generates proton gradient due to proton pumping by the MRC complexes I, III and IV from the matrix to the intermembrane space (IMS). The proton gradient is then used among others to feed ATP synthesis by the ATP synthase. AOX introduces a branch into MRC localized at the ubiquinone/ubiquinol (Q) pool. This results in transferring of electrons to oxygen with sustained proton pumping by the MRC complex I, but without proton pumping by the MRC complexes III and IV, the latter forming together the MRC cytochrome pathway. BHAM, antimycin A and KCN are known inhibitors of AOX, the MRC complex III and IV, respectively.

factors (e.g. [3,7]). Moreover, it is suggested that AOX can bypass blockage or deficiency of the MRC complexes III and IV by restoring electron flow upstream of the MRC complex III. Consequently, AOX is considered to be an important element of a therapeutic strategy against impairment of these complexes [1,3,9–12].

At present, genomes of about 160 animal species representing 16 phyla, from Placozoa to Chordata with exception of insects, lancelets and vertebrates, are proposed to contain AOX encoding genes [1,12–14]. However, the enzyme functionality was only tested in the case of a few species. The best known examples are AOXs of the pacific oyster *Crassostrea gigas* and the tunicate *Ciona intestinalis*. The first one was studied in isolated *C. gigas* mitochondria [15] and after heterologous expression in the yeast *Saccharomyces cerevisiae* cells [16]. The second one was heterologously expressed in mitochondria of cultured human cells [9,17,18] as well as in the fruit fly *Drosophila (Sophophora) melanogaster* [19–23] and mouse [9,12,24,25]. Additionally, some experimental data concerning AOX protein are available for crustaceans, the brine shrimp *Artemia franciscana* and the white shrimp *Litopenaeus vannamei* [26] and the marine copepod *Tigriopus californicus* [27]. In the case of both the shrimps the data concerns AOX activity in isolated mitochondria whereas in the case of the copepod amounts of AOX mRNA and protein in various life stages of the animal and under stress temperature conditions were studied [26,27].

As it has been summarized by Rajendran et al. [9], *C. intestinalis* AOX expressed in human cells, fruit flies and mice is not active when the MRC complex III and/or IV work properly, but inhibition or overload of these complexes triggers the enzyme activity. In the case of human cell lines, *C. intestinalis* AOX expression was shown to confer spectacular cyanide resistance of mitochondrial respiration and compensate for both the growth defect and the pronounced oxidantsensitivity caused by the MRC complex IV deficiency [17,18]. The possibility to provide a complete or substantial protection against a range of phenotypes induced by the MRC complex IV deficiency or inhibition was also observed after expression of *C. intestinalis* AOX in mitochondria of fruit flies [19–21] and mice [24,25]. The same applies to the mouse MRC complex III deficiency [9,11].

Thus, the available data indicates that functional studies of animal AOX are based on models that do not contain native AOX with exception of *T. californicus*, *C. gigas*, *A. franciscana* and *L. vannamei* enzyme. In the case of *C. gigas*, AOX contribution to the oyster adjustments to short-term hypoxia and re-oxygenation was considered in isolated mitochondria [15], but the presented data do not allow for clear conclusions as the observed AA-resistant respiration was not shown to be eliminated by AOX inhibitor, for example by benzohydroxamic acid (BHAM) [28]. Moreover, to observe the enzyme activity in the shrimp isolated mitochondria the authors applied an uncoupler to increase the recorded oxygen uptake rate [26]. Resultantly, the measurements were performed for uncoupled mitochondria, i.e. in the absence of ubiquinone reduction to ubiquinol, regarded as crucial for AOX activity [9,12,20,25]. Thus, data concerning animal AOX expression and activity regulation are still scarce (e.g. [1,3,27]).

In accordance with recent sequencing data, putative AOX encoding genes have been also identified in tardigrades [1,29]. This concerns three species, namely *Hypsibius exemplaris* (former *Hypsibius dujardini*), *Ramazzottius varieornatus* and *Milensium inceptum* (former *Milnesium tardigradum*). Tardigrades are microscopic invertebrates with a body length ranging from 50 μm to 1200 μm although very few species exceed 800 μm. They inhabit marine, freshwater and terrestrial habitats, but all are regarded to be aquatic animals because they require a water-film surrounding the body to be active. The most known feature of tardigrades is the ability to enter cryptobiosis when environmental conditions are unfavorable (for review see for example [30–35]).

Here, we set out to test whether AOX activity of *H. exemplaris*, as a part of MRC, can be observed at the level of intact organisms. We also estimated whether the activity requires the MRC cytochrome pathway blockage. To that end, we determined the animal viability and respiration as well as the mitochondrial inner membrane potential ($\Delta\psi$). The studies were performed in the presence or absence of inhibitors of the MRC cytochrome pathway complexes and AOX. Moreover, we verified the AOX protein presence. The obtained data points at an emergence of a whole-animal model suitable to study activity and expression regulation of natively expressed animal AOX. According to our best knowledge we demonstrated, for the first time, that AOX activity of small aquatic invertebrates can be monitored by measurement of whole-body respiration due to registration of the oxygen consumption rate by intact specimens' mitochondria as well as by detection of $\Delta\psi$ due to application of specific fluorescent dye. Moreover, we demonstrated that the enzyme contributed to animal functioning also in the absence of the MRC cytochrome pathway blockage previously described as precondition to observe animal AOX activity.

## Results

### The AOX inhibitor affects H. exemplaris activity and eliminates the animal tolerance to the MRC cytochrome pathway inhibition

To check functionality of the hypothetical *H. exemplaris* AOX protein (stored in the GenBank under accession number OWA52662.1), we performed *in vivo* toxicity test. We assumed that if *H. exemplaris* specimens had functional AOX, they would tolerate the presence of KCN and BHAM added separately but would be affected by the combination of KCN+BHAM. Additionally, the effect of AA added separately or in the presence of KCN was tested to check the effectiveness of the MRC cytochrome pathway inhibition by KCN. The graphical summary of the test is shown in Fig 2 (KCN and BHAM) and Additional file 1 (includes all the additional controls for the effect of KCN and BHAM including AA). The figures also contain images that illustrate the treated animals showing changes in their body shape and mobility. The animal full activity was defined as coordinated movements of the body and legs. Since the animal treatment with AA in the presence of KCN did not increase the effect of KCN added separately (Additional file 1), we concluded that addition of KCN caused efficient inhibition of the MRC cytochrome pathway.

As shown in Fig 2, in the presence of BHAM specimens initially did not change their mobility and shape, but after 45 min their activity was moderately reduced to frequent leg movements and the animals adopted a croissant shape. The delayed changes triggered by the presence of BHAM were not reversed till the end of the test performed for 2 h. Addition of KCN almost instantaneously and significantly reduced the animal mobility that was confined to single leg movements, however the animals were not paralyzed. Moreover, the body shape of the animals was changed into the characteristic croissant shape. Nevertheless, the animals' activity was partially restored over time within the 2 h of observation. In the presence of KCN and BHAM added in either order, the mobility of *H. exemplaris* specimens was completely stopped within 1 minute and the animals took the characteristic stretched/inflated shape. Furthermore, their return to activity was not detected within the 2 h of observation. Representative images and videos illustrating the appearance and movements of animals in different parts of the toxicity test are presented in Additional files 1 and 2. The obtained results suggested functionality of AOX in *H. exemplaris* mitochondria providing the animal tolerance to the MRC cytochrome pathway inhibition. The inhibition appeared to enhance the AOX activity in a time-dependent way. On the other hand, the cytochrome pathway blockage was not indispensable for observation of AOX contribution to animal functioning.

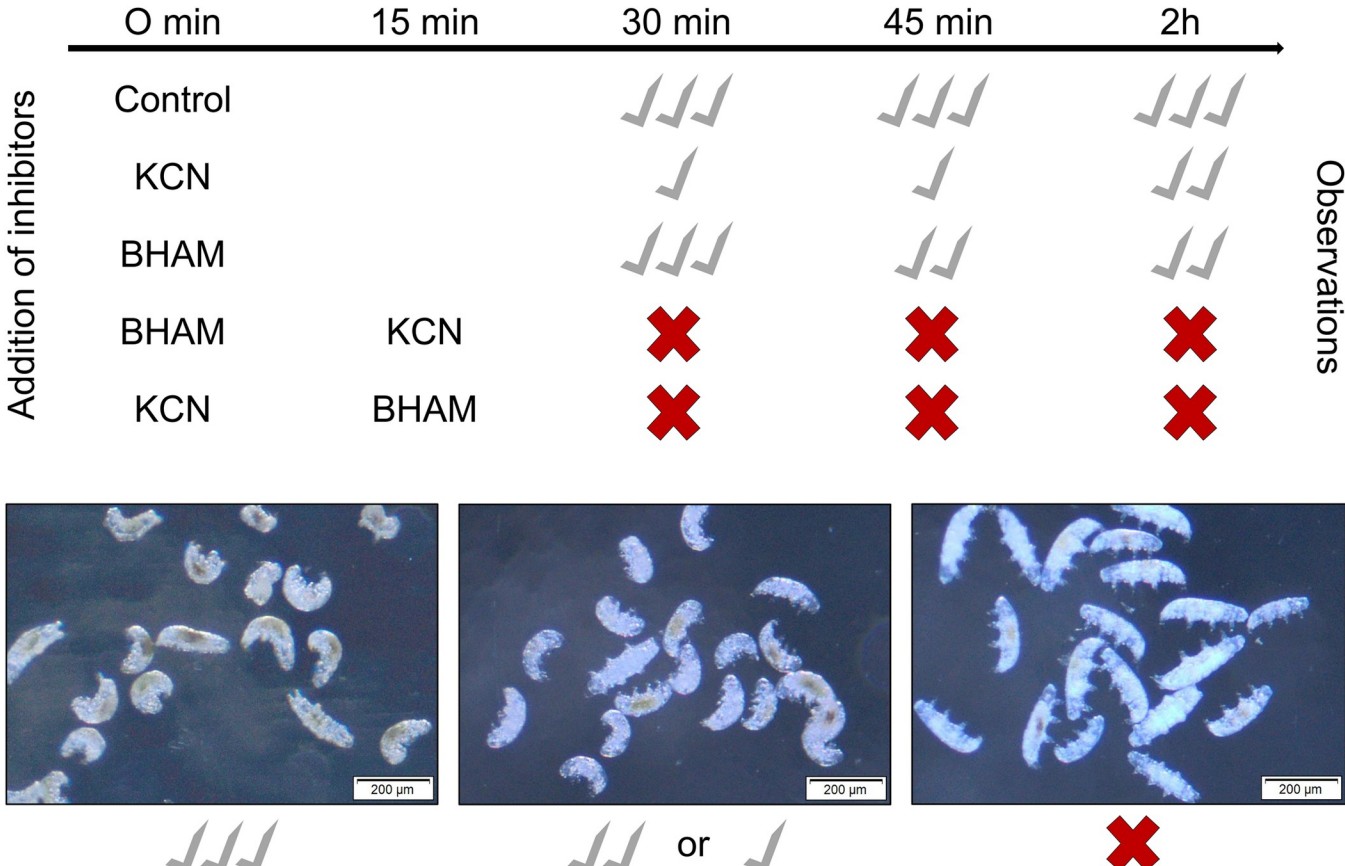

**Fig 2. Toxicity *in vivo* test indicates AOX contribution to functioning of *H. exemplaris* specimens in the absence and in the MRC cytochrome pathway inhibition.** Adult active specimens of a comparable body length and cleaned of debris were treated with KCN (1 mM) and BHAM (3 mM) in different configurations. Animals were observed after 30 min, 45 min and 2 h, and the medium was not replaced till the end of the test. The upper and lower parts of the figure provide graphic representations and images of the treated animals' appearance indicating their body shape changes co-occurring with mobility changes, respectively. The data represent two independent repeats of the test lasting for 2 h and each tested group consisted of 20 specimens (see also Additional file 1 for the performed controls for the MRC cytochrome pathway inhibition by KCN and applied solvents as well as images of animals during and at the end of the test; see also Additional file 2 for recorded films presenting animals after 45 min of the treatment).

### The AOX inhibitor-sensitive respiration enables by-pass of the MRC cytochrome pathway inhibition in H. exemplaris mitochondria but can be also observed in the absence of the inhibition

To verify AOX activity contribution to *H. exemplaris* tolerance to the MRC cytochrome pathway inhibition, we measured the rate of oxygen consumption by intact specimens under the conditions corresponding to the *in vivo* toxicity test. Mitochondria-based respiration of intact specimens was easily measured and affected by the inhibitors of the MRC cytochrome pathway and/or AOX (Fig 3 and Additional file 3 for data on the performed traces). The combination

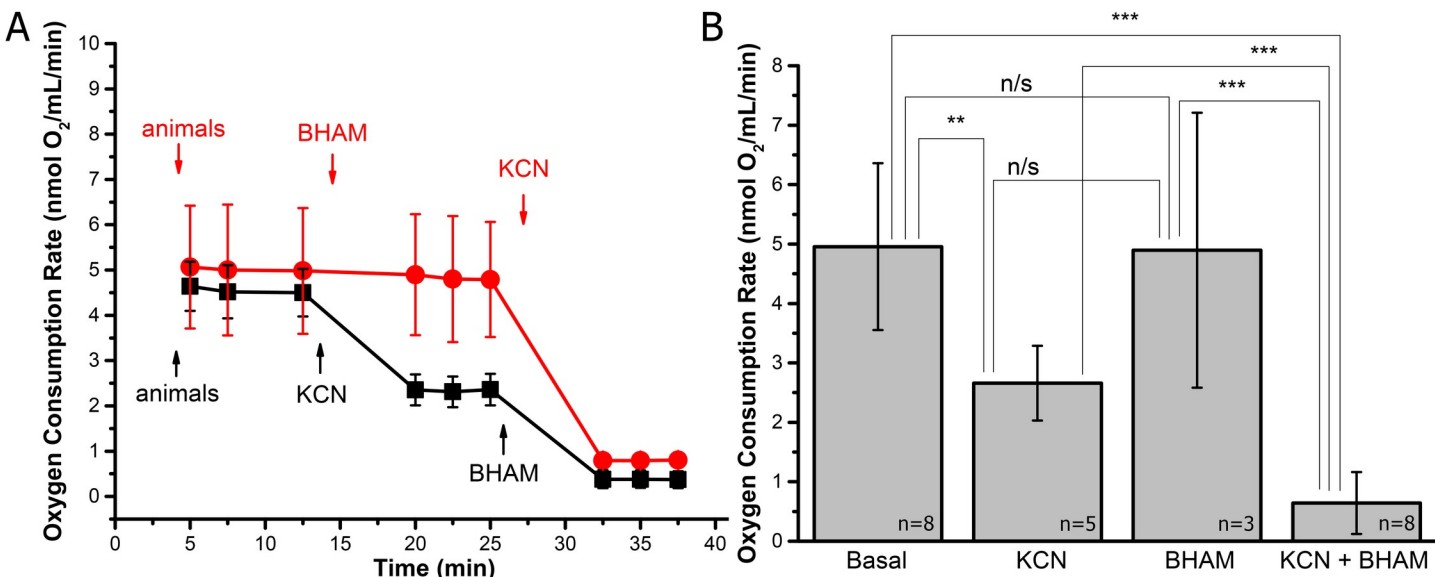

**Fig 3. Measurements of the rate of oxygen consumption by intact *H. exemplaris* specimens confirm the tardigrade AOX activity.** (A) The oxygen consumption rate traces when BHAM was added before or after KCN, determined using Hansatech Oxygraph Plus system. (B) Differences in the oxygen consumption rates calculated for data shown in (A). Basal denotes the oxygen consumption rate after addition of animals. The applied concentrations of inhibitors were as follows 3 mM BHAM and 1 mM KCN. Individual traces were recorded for 10 000 specimens. The statistical significance of results was tested using unpaired t-test for $n \geq 3$. ** $p < 0.01$; *** $p < 0.001$; n/s, not statistically significant (see also Additional file 3 for data on the performed traces including AA addition as control for the MRC cytochrome pathway inhibition by KCN).

of KCN with BHAM caused the respiration elimination independently of the presence of AA (Additional file 3).

Addition of KCN resulted in a decrease of the oxygen consumption by intact specimens and the resulting inhibition was close to 45% (Fig 3A and 3B, respectively). The remaining respiration was sensitive to AOX inhibitor (BHAM) that caused total inhibition of respiration. In the absence of KCN, the recorded respiration was slightly sensitive to BHAM (Fig 3A) but the effect was not statistically significant (Fig 3B) at least for the applied time window (i.e. 12.5 min). However, the inhibition of AOX was effective because the subsequent addition of KCN resulted in total inhibition of the oxygen consumption. It denotes that in the presence of both the inhibitors and within the applied time windows, total inhibition of animals' respiration was observed, independently of the order of KCN and BHAM addition. Moreover, the BHAM-sensitive oxygen consumption by intact specimens could be observed without the blockage of the MRC cytochrome pathway but the blockage was required for the respiration to be pronounced. Accordingly, mass spectrometry allowed for detection of AOX protein (see Additional file 4 for the detection and identification data) only in the protein extract prepared from specimens treated with KCN. The protein extraction was performed at the beginning and at the end of the toxicity test, i.e. after 2 h of treatment with KCN that implies a role of the MRC cytochrome pathway inhibition in modulation of *H. exemplaris* AOX amount. It should be also mentioned that the AOX protein was detected in MW range of 35–40 kDa which corresponds to heterologously expressed animal AOX proteins [19].

## The AOX and the MRC cytochrome pathway inhibitors impose additive effect on the mitochondrial inner membrane potential in H. exemplaris

To further verify data on AOX contribution to *H. exemplaris* functioning, we estimated the mitochondrial inner membrane potential (Δψ) in intact specimens under the conditions

resembling the *in vivo* toxicity test. For that purpose we applied Δψ-sensitive fluorescence dye (TMRM) and calculated the TMRM Fluorescence Index ($FI_{TMRM}$) corresponding to Δψ level. However, it should be mentioned that before fluorescence measurements the applied inhibitors were removed by multiple washing of the treated animals to avoid interference with measurements. As shown in Fig 4A, differences in the fluorescence intensity were observed in animals treated for 45 min with KCN and BHAM added separately or simultaneously. These differences correlated with differences in $FI_{TMRM}$ values calculated for the animal treatment (Fig 4B). Both the treatment with KCN and BHAM resulted in decrease in $FI_{TMRM}$ value, the latter being statistically significant, but application of KCN together with BHAM had additive effect when compared to the inhibitor separate addition. Thus, the changes in animal activity observed after treatment with the MRC cytochrome pathway and AOX inhibitors were also caused by relevant changes in Δψ.

## Discussion

Here, we report that the tardigrade *H. exemplaris* possesses functional AOX that contributes to specimen tolerance to the mitochondrial respiratory chain (MRC) cytochrome pathway inhibition. Simultaneously, according to our best knowledge, it is the first report showing that small aquatic invertebrate AOX activity can be monitored by measurement of intact specimens' whole-body respiration and detection of the mitochondrial inner membrane potential in intact specimens. These could be achieved by measurement of the oxygen consumption rate and intact specimens' staining by Δψ-sensitive fluorescence dye (e.g. TMRM). This makes the tardigrade amenable to whole-organism analyses. So far two other animal models, both obtained due to heterologous expression of the tunicate *C. intestinalis* AOX in fruit flies and mice, have been used in research on AOX impact on animal physiology and the enzyme possible application in therapy of human mitochondrial diseases (e.g. [1,3,9–12]. The tardigrade *H. exemplaris* could supplement the models due to the presence of native AOX and possibility of AOX research in intact specimens. However, it should be remembered that this species is parthenogenetic which could make classical genetic analysis difficult. Nevertheless, availability of the genomic data of *H. exemplaris* at NCBI data base enables studies of targeted genes by RNA interference (RNAi) [36]. Moreover, it is quite possible that as in the case of RNAi approach, CRISPR/Cas9 approach adapted from protocol originally developed for *Caenorhabditis elegans* will be soon obtained. This in turn will allow manipulation with AOX expression and/or introduction of mutations of the MRC cytochrome pathway proteins resulting in mitochondrial dysfunctions mimicking the mitochondrial respiratory chain impairments observed in human diseases. Accordingly, fast reproduction and simple culture protocol [e.g. 37,38] as well as not sophisticated procedure of monitoring mitochondria functioning in intact animals including the whole-body respiration and mitochondria coupling, could enable analysis of the affected animals' physiology and facilitate fast screening of pharmacologically active compounds. Moreover, *H. exemplaris* might provide valuable data concerning AOX involvement in efficient protection against stress factors affecting mitochondria [39]. This also applies to aging as the mean lifespan under laboratory conditions of about two months makes the species suitable for studies of AOX activity role in aging. Thus, *H. exemplaris* may constitute a genetically tractable model enabling monitoring of functional consequences of the natively expressed AOX activity and the level of the protein expression itself. As we have recently shown for the tardigrade *M. inceptum*, BHAM can be described as an efficient inhibitor of tardigrade AOX [39].

The common property of *C. intestinalis* AOX heterologously expressed in fruit fly and mouse mitochondria is that it confers a substantial tolerance to the MRC cytochrome pathway inhibitors *in vivo* [23,28,29] but becomes enzymatically active only when MRC becomes

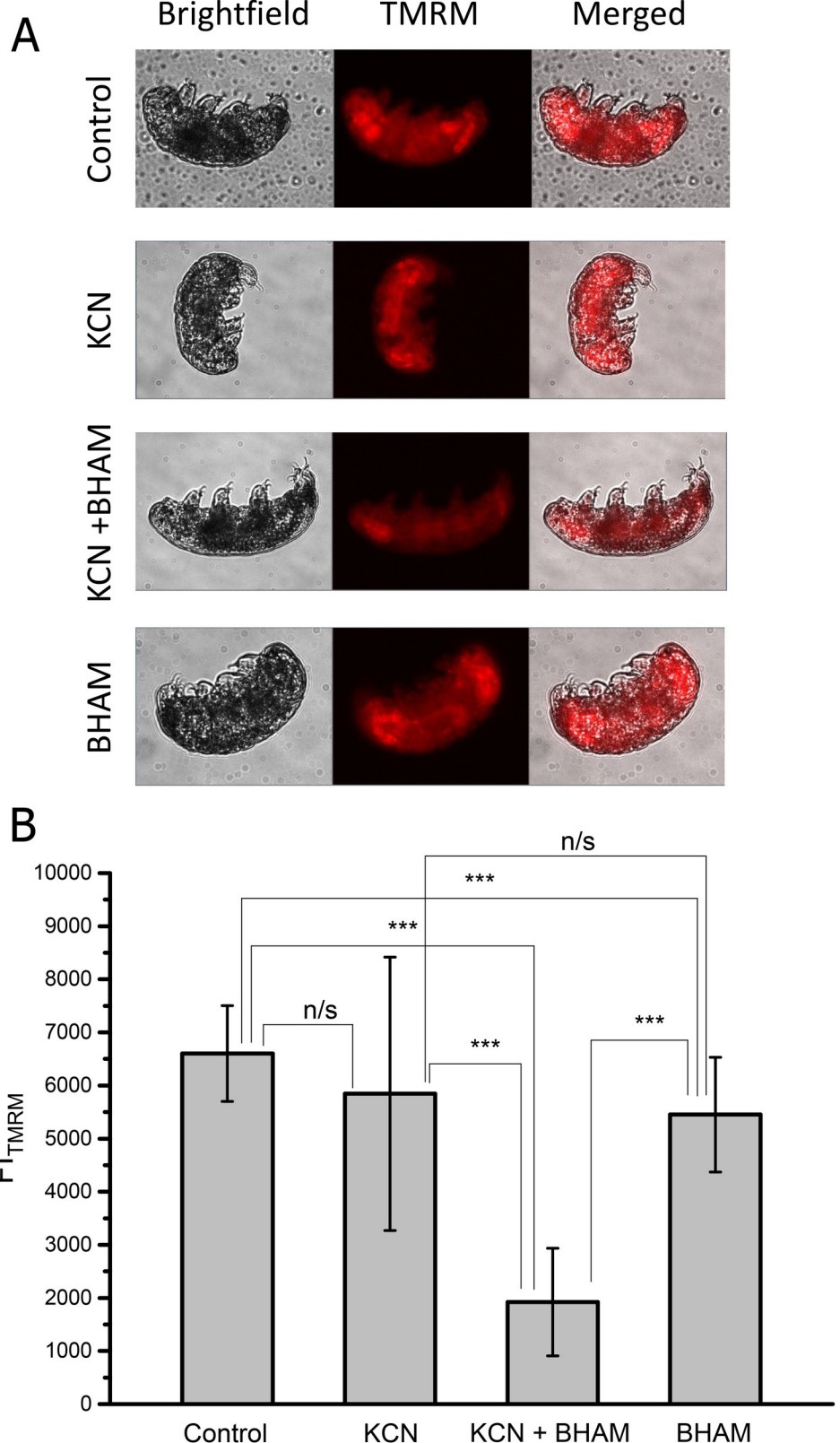

**Fig 4. The measurements of the mitochondrial inner membrane potential (Δψ) in intact *H. exemplaris* specimens confirm the tardigrade AOX activity.** (A) Representative fluorescence microscopic images of animals treated with

KCN and BHAM applied separately or together in the absence of FCCP. (B) Quantification of TMRM signals obtained for animals treated with KCN and BHAM applied separately or together and subsequently treated with FCCP. The FI$_{TMRM}$ represents the value of the net transmembrane potential and defined as the value of the TMRM fluorescence level under a given condition eliminated in the presence of FCCP. The applied concentrations of compounds were as follows: 2 μM TMRM, 3 mM BHAM, 1 mM KCN and 10 μM FCCP. The statistical significance of results was tested using unpaired t-test for n = 24. *** $p < 0.001$; n/s, not statistically significant.

inhibited beyond ubiquinone [9,17,18,20]. In the case of natively expressed *H. exemplaris* AOX, the specimens' distinct BHAM-sensitive respiration was observed in the presence of previously added KCN, but longer treatment with BHAM resulted in Δψ decrease that implies importance of the time factor in the occurrence of AOX inhibition effect in the absence of the MRC cytochrome pathway inhibition. Accordingly, *H. exemplaris* specimens treated with BHAM in the absence of the MRC cytochrome pathway inhibition, initially did not change their mobility and shape, but after 45 min of the treatment their activity was moderately reduced, and their shape changed. This appears to corroborate to slight inhibition of the oxygen consumption observed in the presence of BHAM within the treatment time shorter than 45 min that could denote a need of adaptation to AOX inhibition. AOX is not a proton pump, hence the inhibition of its activity by BHAM should not affect Δψ when the main complexes of the respiratory chain are active. This observation requires further studies as till now the effect of prolonged inhibition of animal AOX has not been studied. However, partial inhibition of animal respiration imposed by KCN did not result in a distinct Δψ decrease suggesting that under these conditions active *H. exemplaris* AOX is able to maintain Δψ by cooperating with the proton pumping complex I [40]. This stays in agreement with the observation obtained for isolated mouse heart mitochondria with heterologous *C. intestinalis* AOX expression that AOX cooperates with complex I only in the presence of the cytochrome pathway inhibition [12].

On the other hand, specimens treated with KCN, after initial restrained mobility, partially returned to activity at the end of the toxicity test, i.e. after 2 h and this return was prevented by the presence of BHAM. This, in turn, could be explained as a consequence of AOX increasing role in the presence of the MRC cytochrome pathway inhibition. Accordingly, *H. exemplaris* AOX protein detection was possible only after specimens treatment with KCN. The blockage of the MRC cytochrome pathway results in hypoxia or even anoxia described as important factors controlling the animal AOX transcript levels [3,17,41]. Moreover, AOX affinity to its quinone substrate may increase under condition of the substrate prolonged reduction resulting from the blockage of the MRC complexes III and/or IV (e.g. [23]). Nevertheless, the obtained results suggest that *H. exemplaris* specimens are probably more tolerant to KCN treatment than AOX-expressing fruit flies. The latter remained active during 20–30 min of the KCN treatment and then recovered from paralysis overnight [19]. It has also been shown that the heterologous AOX expression in mice enables the anesthetized animal survival after exposition to a lethal concentration of gaseous cyanide [24] or injected KCN [25]. It is clear that without the use of anesthesia, these experiments would have been inhumane although it should be remembered that anesthetics may contribute to mitochondrial dysfunction [24,42].

Summing up, the activity of *H. exemplaris* AOX in intact specimens can be observed by monitoring oxygen consumption and Δψ changes. The activity is affected by the presence of the MRC cytochrome pathway inhibition and the impact of the activity elimination by application of AOX inhibitor seems to be time-dependent. Moreover, the amount of AOX in *H. exemplaris* specimens increases distinctly, above the threshold of detection by mass spectrometry, when these animals cope with the MRC cytochrome pathway inhibition. This indicates at important mechanism of the encoding gene regulation at protein level imposed by the

blockage of the MRC cytochrome pathway inhibition that cannot be fully studied using animals with the AOX heterologous expression controlled in an artificial way.

## Conclusions

As has been mentioned by Saari et al. [3], despite the encouraging findings to date in animal models, the potential problems in the use of AOX in human disease therapy need to be considered in much greater detail. Accordingly, application of *H. exemplaris*, could help to explain inherent properties of natively expressed AOX, in particular the encoding gene expression and the enzyme activity regulation, and how the activity affects the organism. Thus, we propose consideration of the tardigrade *H. exemplaris* application as a whole-organism animal model for research on natively expressed AOX that enables analysis of the enzyme contribution to animal behavior and mitochondria functionality under both physiological conditions and a given biochemical defect or external stress. It is also important that protocols for *H. exemplaris* gene expression and genetic manipulation were developed (e.g. [36,43]) that increases the range of possible experiments, especially that this species is used as a model species in other research.

## Materials and methods

### Reagents

The following inhibitors were applied: benzohydroxamic acid (BHAM; #412260) for AOX, potassium cyanide (KCN; #60178) for the MRC complex IV and antimycin A (AA; #A8674) for the MRC complex III (purchased from Sigma-Aldrich). For the detection of the mitochondrial inner membrane potential tetramethylrhodamine methyl ester (TMRM; ThermoFisher T668) was applied. Carbonyl cyanide 4-(trifluoromethoxy)phenylhydrazone (FCCP; # 2920) was used to eliminate the mitochondrial inner membrane potential.

### Culture of Hypsibius exemplaris

Z151 strain of the parthenogenetic species *Hypsibius exemplaris* was purchased from Sciento (Manchester, United Kingdom) and cultured according to the published protocol [37]. To maintain the culture, specimens were kept in POL EKO KK 115 TOP$^+$ climate chamber (photoperiod 12h light/12h dark, 20 $^o$C, relative humidity of 50%) on Petri-dishes (55 mm in diameter) with bottom scratched by sandpaper to allow tardigrade locomotion. They were covered with a thin layer of the culture medium obtained by mixing double-distilled water and spring water (Żywiec Zdrój S.A., Poland; bicarbonates: 121.06 mg/L, fluorides: 0.07 mg/L, Mg$^{2+}$: 5.37 mg/L, Ca$^{2+}$: 36.39 mg/L, Na$^+$: 7.79 mg/L) in ratio of 3 to 1. *Chlorella vulgaris* SAG211-11b strain was served as a food once per week after the plate cleaning. Animals were transferred to new culture plates every few months as they number double every two weeks. The SAG211-11b strain was a kind gift of Marcin Dziuba (Department of Hydrobiology, Faculty of Biology, Adam Mickiewicz University, Poznań, Poland) and was obtained from the culture collection of algae (Sammlung von Algenkulturen (SAG)) at the University of Göttingen, Germany.

### Toxicity in vivo test

The test was performed in glass blocks with hemispherical cavity of 32 mm in diameter (Karl Hecht Staining Blocks Molded Glass 2020, Lab Unlimited). Only adult and fully active (displaying coordinated movements of the body and legs) specimens with a body length of about 200 μm and cleaned of debris were collected 16 h before the test beginning and kept in the glass block in 3 ml of the culture medium. To check the impact of the MRC cytochrome

pathway and/or AOX inhibition, groups of 20 fully active specimens were transferred into 1 ml of the culture medium in new glass blocks. Then these animals were treated with KCN, BHAM, Methanol and AA added separately or in combinations. AA was used as a control for the effective MRC cytochrome pathway inhibition by KCN and methanol was used as a solvent control for AA and BHAM (Additional file 1). The treated animals were kept at 18 °C and observed after 30 min, 45 min and 2 h. The medium was not replaced till the end of the test. The applied concentrations of KCN, AA and BHAM were established experimentally to obtain saturated effect on the oxygen consumption rate inhibition (see Measurements of *H. exemplaris* respiration) and were as follows: 1 mM KCN, 3 mM BHAM and 280 μg/ml AA. The test lasted two hours and was repeated two times. Specimens activity before and after the treatment was monitored under Olympus SZ61 stereomicroscope (at 45 x magnification) connected to Olympus UC30 microscope digital camera. Images and short video films were obtained using Olympus CellSens Standard Software.

## Measurements of H. exemplaris respiration

To estimate animal respiration, the rate of oxygen consumption was measured at 18°C in 0.5 ml of the culture medium, using two water-thermostated incubation chambers with computer-controlled Clark-type $O_2$ electrode (Oxygraph, Hansatech, UK). The adult fully active specimens (about 200 μm in a body length) were collected as described above for toxicity *in vivo* test in the amount of maximally 2000 specimens per each glass block. To obtain specimens with empty gut, (to avoid an impact of AOX belonging to *C. vulgaris* applied as a food) they were starved for three days and the culture medium was exchanged once per day. Then, the specimens were transferred to a glass test-tube, and kept in 10 ml of the culture medium. The medium was replaced 3–4 times before the measurements to avoid hypoxia. A trace was recorded for a suspension of 10 000 *H. exemplaris* specimens (without algae in the gut) in 0.1 ml of the glass test-tube medium. The proper control trace was performed simultaneously in the second chamber in the absence of animals, but with addition of 0.1 ml of the glass test-tube medium and the applied reagents. To avoid hypoxia, the oxygen consumption rate measurements were finished when oxygen level achieved 50% of its maximal value. The measurements were performed at least in triplicate. The statistical significance of results was tested using unpaired t-test.

## Estimation of the mitochondrial inner membrane potential in H. exemplaris specimens

The adult fully active specimens were collected and starved as described above for Measurements of *H. exemplaris* respiration. Then, 10 000 animals were transferred into 2 ml of the culture medium in a new glass block and treated with 2 μM TMRM for 1 hour at 18°C. Next, the treated animals were washed carefully by multiple replacement of the culture medium and subsequently were kept for 1 hour at 18°C in the fresh culture medium to remove excess of the dye. Then, 500 animals were transferred per each glass block containing 1 ml of the culture medium. These animals were treated with 1 mM KCN or 3 mM BHAM or 1 mM KCN and 3 mM BHAM added together. The treated animals were kept for 45 min at 18 °C and then washed carefully by multiple replacement of the culture medium. After the washing the treated animals were transferred in 100 μl of the culture medium into multi-well plate (Corning #3573) and three independent fluorescence measurements octuplicated by multiple reading per well were made immediately at excitation wavelength 544 nm and emission wavelength 590 nm using Tecan infinite 200Pro microplate reader. After the first series of measurement, the plate was removed from the reader and 10 μM FCCP was added to each well and incubated

45 min. Then the second series of the measurement was performed. The data were exported by Tecan i-control software and recalculated to the TMRM Fluorescence Index ($FI_{TMRM}$), representing the value of the net transmembrane potential and defined as the value of the TMRM fluorescence level under a given condition eliminated in the presence of FCCP. The statistical significance of results was tested using unpaired t-test. The animals stained with TMRM and treated with KCN and BHAM were observed under Axio Observer.Z1 (Carl Zeiss, Jena, Germany) inverted fluorescence microscope (at 160 x magnification) connected to digital camera. Images were obtained using ZEN 3.3 (blue edition) software.

## Detection of AOX protein

The presence of AOX protein was estimated by liquid chromatography coupled to tandem mass spectrometry (LC–MS/MS) performed in the Department of Biomedical Proteomics, Institute of Bioorganic Chemistry, Polish Academy of Sciences (Poznań, Poland). Amino acid sequences annotated as animal AOX and stored in the GenBank were used for the presence detection. The analyzed samples were cut from SDS-PAGE gel at its part corresponding to MW between 25–34 and 35–40 kDa. Each sample was obtained for 15 000 adult fully active specimens with empty gut incubated at 18 $^{\circ}$C in 1 ml of the culture medium in the glass cubes in the absence or in the presence of 1 mM KCN at the beginning and at the end of the toxicity test (i.e. after 2 h of the treatment). Then animals were pelleted in 1.5 ml Eppendorf tubes (8 000 g for 2 min), and the pellet was frozen in liquid nitrogen and stored at -80$^{\circ}$C until protein extraction. To obtain the extract, 50 µl of an isolation buffer (50 mM ammonium bicarbonate, 1% SDS, 1x Protein Inhibitor Thermo Fisher Scientific, #78430) were added to each tube containing the pellet. Then, the samples were frozen and thawed three times, sonicated using Bioruptor Plus UCD-300 (4$^{\circ}$C, 20 kHz, 15 x 30 s. with 30 s interval between each sonication pulse) and centrifuged (8 000 g for 2 min in 4$^{\circ}$C). The obtained supernatant (app. 50 µl) was transferred from each Eppendorf tube to the new one. Total amount of proteins in each of the supernatants was estimated by Bradford assay (5 replications). The 60 µg of each sample was denaturated using Sigma 2x lysis buffer (#3401) for 5 min in 95$^{\circ}$C and loaded on SDS-PAGE (Sodium Dodecyl Sulfate-Polyacrylamide Gel Electrophoresis) gel. Protein separation in 14% resolving gel was performed according to standard protocol [44].

## Supporting information

**S1 Fig. Images of the treated animals' activity indicating their body shape changes co-occurring with mobility changes after 30 min. and 2 h of the toxicity *in vivo* test including controls for the applied inhibitor solvents and the MRC cytochrome pathway inhibition by KCN.** The data represent two independent repeats of the test and each tested group consisted of 20 specimens. The images present all variants of the performed test.
(PDF)

**S1 Table. Data on the performed traces for mitochondria-based respiration of intact specimens including AA addition as a control for the MRC cytochrome pathway inhibition by KCN.**
(DOCX)

**S2 Table. LC-MS/MS based detection of *H. exemplaris* AOX protein supplemented by comparison of the *H. exemplaris* AOX predicted amino acid sequence (OWA52662.1) with the sequence of *Ciona intestinalis* [45].** The sequence alignment was obtained by ClustalW Multiple Alignment tool [46], implemented in BioEdit. The identified peptides are marked with color boxes. The conserved glutamate (E) and histidine (H) residues within the ferritin-like

domain (black frame) are marked with stars.
(DOCX)

**S1 Movie. Recorded films presenting animals after 45 min treatment with BHAM, KCN and BHAM + KCN as well as without the treatment.**
(MP4)

## Acknowledgments

We are extremely grateful to Magdalena Łuczak (Department of Biomedical Proteomics, Institute of Bioorganic Chemistry, Polish Academy of Sciences, Poznan, Poland) for her invaluable contribution to LC–MS/MS analysis. We would also like to thank to Sławek Cerbin and Marcin K. Dziuba for *Chlorella vulgaris* SAG211-11b strain used to feed the cultured *H. exemplaris* specimens. Studies have been partially conducted in the framework of activities of BARg (Biodiversity and Astrobiology Research group).

## Author Contributions

**Conceptualization:** Daria Wojciechowska, Łukasz Kaczmarek, Wiesława Jarmuszkiewicz, Hanna Kmita.

**Formal analysis:** Daria Wojciechowska, Milena Roszkowska, Andonis Karachitos.

**Funding acquisition:** Hanna Kmita.

**Investigation:** Daria Wojciechowska, Milena Roszkowska, Andonis Karachitos.

**Methodology:** Daria Wojciechowska, Milena Roszkowska, Andonis Karachitos.

**Project administration:** Hanna Kmita.

**Resources:** Milena Roszkowska.

**Supervision:** Łukasz Kaczmarek, Hanna Kmita.

**Validation:** Daria Wojciechowska, Milena Roszkowska, Łukasz Kaczmarek, Wiesława Jarmuszkiewicz, Andonis Karachitos, Hanna Kmita.

**Visualization:** Milena Roszkowska, Andonis Karachitos, Hanna Kmita.

**Writing – original draft:** Daria Wojciechowska, Milena Roszkowska, Łukasz Kaczmarek, Wiesława Jarmuszkiewicz, Andonis Karachitos, Hanna Kmita.

**Writing – review & editing:** Daria Wojciechowska, Milena Roszkowska, Łukasz Kaczmarek, Wiesława Jarmuszkiewicz, Andonis Karachitos, Hanna Kmita.

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
