## [Decision Letter · Decision Letter 0]

27 Jan 2021

PONE-D-20-38224

The tardigrade Hypsibius exemplaris as an emerging model to study the mitochondrial alternative oxidase at animal organismal level

PLOS ONE

Dear Dr. Kaczmarek,

Thank you for submitting your manuscript to PLOS ONE. After careful consideration, we feel that it has merit but does not fully meet PLOS ONE’s publication criteria as it currently stands. Therefore, we invite you to submit a revised version of the manuscript that addresses the points raised during the review process.

Personally, I find this study very interesting. However, the data presented is very limiting to assess the impact of AOX on overall mitochondrial bioenergetics of H. *exemplaris*. I would prefer to see the followings:

The effects of different inhibitors including BHAM and other AOX inhibitors (if any, prolyl gallate?) in the presence and absence of AA and KCN on ATP and mitochondrial membrane potential.Molecular identification and characterization of the AOX protein/gene from *H. exemplaris. *Modification of the text as suggested by the reviewers.

We look forward to receiving your revised manuscript.

Kind regards,

Nagendra Yadava, Ph.D.

Academic Editor

PLOS ONE

Journal Requirements:

2. Please amend the manuscript submission data (via Edit Submission) to include author Daria Grobys.

3. Please amend your authorship list in your manuscript file to include author Daria Wojciechowska.

Reviewers' comments:

Reviewer's Responses to Questions

**Comments to the Author**

1. Is the manuscript technically sound, and do the data support the conclusions?

Reviewer #1: Yes

Reviewer #2: Yes

2. Has the statistical analysis been performed appropriately and rigorously? 

Reviewer #1: Yes

Reviewer #2: No

3. Have the authors made all data underlying the findings in their manuscript fully available?

Reviewer #1: Yes

Reviewer #2: Yes

4. Is the manuscript presented in an intelligible fashion and written in standard English?

Reviewer #1: Yes

Reviewer #2: Yes

5. Review Comments to the Author

Reviewer #1: The paper of Grobys et al., “The tardigrade Hypsibius exemplaris as an emerging model to study the mitochondrial alternative oxidase at animal organismal level” demonstrates functionality of recently revealed ubiquinol oxidase in tardigrade species Hypsibius exemplaris. The work could be especially interesting for comparative physiologists and biochemists that focus on alternative respiratory pathways in animals and the role of these pathways in animal adaptation to functional hypoxia, respiratory chain inhibitors, temperature changes, and other stressors.

The work is quite laconic, however the results presented seem trustable and complete according to the authors’ intentions.

I do not have serious concerns regarding the experimental design and methodology as well as data representation. However, it is worth noting a number of minor concerns that should be taken into account to improve the reading and understanding of the authors’ introduction, data, and discussion:

Abstract, line 16:

The authors state “(AOX) is present in mitochondria of many invertebrates”. This sounds too general. In my opinion, it would be better to state “several invertebrate taxa” instead of “many invertebrates”. So far, alternative oxidase (AOX) was revealed in genomes of some sea squirts, crustaceans, mollusks, annelids, cnidarians, and sponges but not insects. It is not known so far whether occurrence of AOX in genomes of the listed invertebrate taxa is due to horizontal gene transfer. If the gene that encodes AOX was horizontally transferred from symbiotic fungi, plants or protists to the animals, then likely not all representatives of a particular taxon may contain it.

Abstract, line 21: The abbreviation MRC is not deciphered. It is present in the List of abbreviations, however direct introduction of this abbreviation in the Abstract would be desirable.

Abstract, lines 26-28: The authors did not study AOX expression (e.g. mRNA levels or detection by western blot, or enzymatic activity with its native substrate, ubiquinol) therefore it is not relevant to write that it was monitored.

Keywords, line 31:

I would suggest substitution the term “intact organism” with something like “whole-body respiration” or “whole organism/animal respiration”, as well as the term “KCN tolerance” with “cyanide resistance”.

Introduction, line 49: “understood” instead of “understand”

Introduction, line 77: In my opinion, there is no need to take additional reference for defining Drosophila melanogaster species since the organism is well-known. The same concern is about references 30, 31, and 33. They contain narrow taxonomical data and do not have relation to the main theme of the article which is considered to fall in the field of comparative physiology.

Introduction, lines 80-82: There should be a reference at the end of the sentence.

Introduction, line 89: I suggest “oxidant sensitivity” instead of “oxidant-sensitivity”.

Introduction, lines 96-98: The sentence is irrelevant. The study of Tward et al. (2019) criticized by the authors does not represent a functional study of the animal AOX but is rather a detection of AOX in the copepode Tigriopus californicus. The study of Tward et al. (2019) suggests the molecular mass of putative AOX basing on the preliminary immunobloting analysis but does not conclusively postulate it. The authors state “... its predicted amino acid sequence does not contain the proper C-terminus motif [34]”. What do they mean under “proper” C-terminus motif? Tward and colleagues have found C-terminal motif NPFEKGK of AOX from T. californicus which is indeed similar to the C-terminal motif of AOXs from other animals, e.g. Amphimedon queenslandica (NPFEPGK), Ciona intestinalis and Phallusia mammillata (NPYPPGQ), Branchiostoma floridae (NPFQPGK), and many others.

Introduction, lines 109-111: The references 37-40 do not have relation to the AOX genes or sequencing projects, therefore are misleading in relation to the context.

Introduction, line 118: Here, the authors use the term “mitochondrial respiratory system” whereas “mitochondrial respiratory chain” is in the abstract. I would suggest to either unify terminology throughout the text or explain the difference between the chain and the system.

Results, lines 165-169, and Figure 3:

The figure would benefit if contained respiration rates near the appropriate segments of the oxygen consumption curve. The marks of statistical significance should be shown on the Figure 3B.

Methods, line 253: A reference to the mineral content of the water would be helpful.

Methods, lines 272-275 or legends for Figures 2, and S1, and Figures 2, and S1:

The magnification of microscope (in the Materials and Methods and/or Figure legends) and scale bars in the corresponding figures would improve perception of the figures for readers.

Methods, lines 300-301: Formally, the abbreviation SDS-PAGE should be deciphered (despite it is widely used and well-known).

Reviewer #2: This paper tries to demonstrate a functional relationship between mitochondrial respiratory chain (MRC) and alternative oxidase (AOX) in the tardigrade Hypsibius exemplaris. Based on the measurements of respiration in the presence and absence of pharmacological inhibitors of the MRC and AOX, it proposes that H. exemplaris can serve as model for the study of mitochondrial AOX. While the presented data indeed indicates the possibility of H. exemplaris being a potential model organism for the study of AOX and its relationship with MRC in whole body physiology, the presented data is insufficient. There are several issues that must be clearly addressed.

1. Describe the criteria for selecting an animal model to study AOX and why H. exemplaris fits these criteria. Just BHAM sensitivity is does not make it a good model to study AOX. Other biological and genetic features must be described and how it can be used to model the consequences of mitochondrial dysfunction and therapeutic strategies for mitochondrial dysfunction in details.

a. It will be good to provide some information about how fast animals double (i.e. reproduction cycle). Otherwise if the animals become limiting, then this model is no good for experimental purposes. Also some information on their genetics and potential genetic tractability and its utility to model human disease/aging will be useful.

2. Figure 1: It would be good to show the stoichiometry of H+ pumped out with and without AOX operating and discuss its impact on the ATP yield.

3. Apart from respiration assay, no attempt was made to understand the impact of AOX and MRC inhibitions on ATP and mitochondrial membrane potential. Measurements of ATP and membrane potential do not require species-specific reagents such as antibody. Therefore, ATP and mitochondrial membrane potential data can be easily obtained using standard procedures with minimal optimizations. Such data are essential to understand the role of AOX in mitochondrial bioenergetics.

4. Evidence for the BHAM sensitivity and its specificity to AOX of H. exemplaris should be provided. BAHM sensitivity and mass spectrometry do not provide direct evidence for AOX function. One reference cited as evidence for the specificity of BHAM for AOX appears to be a review article.

5. Figure 2 is highly confusing. Both figure legends and method sections do not provide enough details of the experiments.

a. All data should be reported in quantitative terms by giving the % animals responding to inhibitor(s) at each time (or one chosen time point). For example % animals mobile or immobile and their shapes at each time point.

b. Data from Figueres 2 and S1 can be combined in one figure. Only the video file should be presented as a supplement.

c. How the reversal of phenotype was tested? By replacing the medium with fresh medium without inhibitors or just wait and watch for 2 hours? If by wait and watch, then why reversal would be expected in the presence of inhibitors.

6. Page 7, lines 153-54: The statement…” the rapid disappearance of full activity was also observed when BHAM was added in the presence of AA applied in a sequence with KCN” is unclear.

a. Was there no synergistic effect of AA with BHAM without KCN? If so, why?

b. From Figure S1, the morphology with AA treatment alone looks like that observed with BHAM.

7. Figure 3:

a) Panel A) rates before and after each treatment should be given in the graph for easy comparison. Visually, BHAM’s effect is not apparent as described in the text.

b) Panel B) Were the data obtained from experiments like in panel A? How many repeats were performed for the reported data in panel B? Is the reported data from one representative experiments or average of multiple experiments? How does it correlate with the data in Table 1? Is the reported data here based on 1st order or 2nd order kinetics?

c) Was the respiratory inhibition by BHAM alone significant? Show asterisks indicating significance level in panel B within the graph.

8. Table S1: Insufficient details is provided to understand this table.

a. Are the data in each row are from same experiment or different experiment? How many animals were used for obtaining these data? What is the respiration rate/animal? What is the degree of inhibition by each inhibitor individually and in combinations? Were the values reported for KCN, AA, and BHAM columns obtained independently or by sequential additions?

b. It would be appropriate to show a graph superimposed with respiratory profile and the rates together for the sake of clarity.

c. Preferably, the respiration rate should be reported as the rate of oxygen consumption rather than the rate of uptake. Even though respiration rate will equal to the rate of uptake in principle, the uptake is not being measured directly. Oxygen consumption would be appropriate term as it is occurring to some degree even in the absence of animals in the chamber.

9. There is a mismatch in the functional data and AOX protein data supposed to detected by mass spectrometry only following 2 h treatment with KCN.

a. No efforts have been made to monitor the AOX protein or clone the gene using the information available to further characterize the AOX protein. It is possible that polyclonal antibody against AOX from other species can detect AOX of H. exemplaris. Alternatively, based on the protein sequence obtained from mass spectrometry and the sequences from other species, it would be possible to clone the gene by degenerate PCR, develop antibodies, compare, and characterize the AOX protein of H. exemplaris.

b. The BHAM sensitivity alone is not enough to inform about H. exemplaris AOX function in mitochondrial bioenergetics. It is interesting to note that BHAM sensitivity is observed within 30-40 min (Fig. 3A) and the APX protein is detected following 2 h KCN treatment even with mass spectrometry, which is considered to be a highly sensitive technology. This discrepancy challenges the very basic foundation of this manuscript – that H. exemplaris AOX is function without the inhibition of Complexes III/IV.

6. PLOS authors have the option to publish the peer review history of their article (what does this mean?). If published, this will include your full peer review and any attached files.

Reviewer #1: No

Reviewer #2: No

---

## [Author Response · Author response to Decision Letter 0]

11 Jul 2021

Dear Editor,

Answers for all reviewers and editor comments are attached as separte file.

Kind Regards,

Łukasz Kaczmarek

---

## [Editor Report · Decision Letter 1]

10 Aug 2021

The tardigrade Hypsibius exemplaris has the active mitochondrial alternative oxidase that could be studied at animal organismal level

PONE-D-20-38224R1

Dear Dr. Kaczmarek,

We’re pleased to inform you that your manuscript has been judged scientifically suitable for publication and will be formally accepted for publication once it meets all outstanding technical requirements.

Kind regards,

Nagendra Yadava, Ph.D.

Academic Editor

PLOS ONE

Additional Editor Comments (optional):

I believe that in the title "...animal or organismal level" should read as "....animal or organismal level". Please correct it during the proof stage or as advised by the PLOS ONE during the publication process.
---

## [Editor Report · Acceptance letter]

13 Aug 2021

PONE-D-20-38224R1 

The tardigrade *Hypsibius exemplaris*  has the active mitochondrial alternative oxidase that could be studied at animal organismal level. 

Dear Dr. Kaczmarek:

I'm pleased to inform you that your manuscript has been deemed suitable for publication in PLOS ONE. Congratulations! Your manuscript is now with our production department. 

Kind regards, 

on behalf of

Dr Nagendra Yadava 

Academic Editor

PLOS ONE